# Psoralidin induces autophagy through ROS generation which inhibits the proliferation of human lung cancer A549 cells

Wenhui Hao, Xuenong Zhang, Wenwen Zhao and Xiuping Chen

State Key Laboratory of Quality Research in Chinese Medicine, Institute of Chinese Medical Sciences, University of Macau, Macao, China

## ABSTRACT

Psoralidin (PSO), a natural furanocoumarin, is isolated from *Psoralea corylifolia L.* possessing anti-cancer properties. However, the mechanisms of its effects remain unclear. Herein, we investigated its anti-proliferative effect and potential approaches of action on human lung cancer A549 cells. Cell proliferation and death were measured by MTT and LDH assay respectively. Apoptosis was detected with Hoechst 33342 staining by fluorescence microscopy, Annexin V-FITC by flow cytometry and Western blot analysis for apoptosis-related proteins. The autophagy was evaluated using MDC staining, immunofluorescence assay and Western blot analyses for LC3-I and LC3-II. In addition, the reactive oxygen species (ROS) generation was measured by DCFH2-DA with flow cytometry. PSO dramatically decreased the cell viabilities in dose- and time-dependent manner. However, no significant change was observed between the control group and the PSO-treated groups in Hoechst 33342 and Annexin V-FITC staining. The expression of apoptosis-related proteins was not altered significantly either. While the MDC-fluorescence intensity and the expression ratio of LC3-II/LC3-I was remarkably increased after PSO treatment. Autophagy inhibitor 3-MA blocked the production of LC3-II and reduced the cytotoxicity in response to PSO. Furthermore, PSO increased intracellular ROS level which was correlated to the elevation of LC3-II. ROS scavenger N-acetyl cysteine pretreatment not only decreased the ROS level, reduced the expression of LC3-II but also reversed PSO induced cytotoxicity. PSO inhibited the proliferation of A549 cells through autophagy but not apoptosis, which was mediated by inducing ROS production.

## INTRODUCTION

*Psoralea corylifolia Leguminosae* (L.), an herb widely distributed in China and Southeastern Asian countries, has been used as a multi-purpose medicinal plant (*Zhao et al., 2005*). The fully mature dried fruit of this plant is well-known as traditional Chinese medicine called "Buguzhi", and is used to treat a wide range of diseases. Previous screening studies have reported that some extracts and active fractions of *P. corylifolia L.* exhibited cytotoxicity and inhibition of chemical carcinogenesis (*Latha et al., 2000*; *Latha & Panikkar, 1999*;

Corresponding author
Xiuping Chen, xpchen@umac.mo

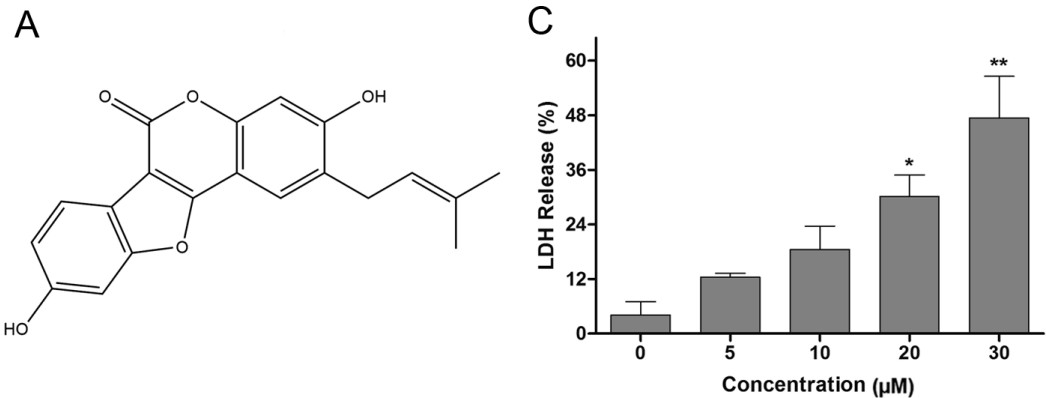

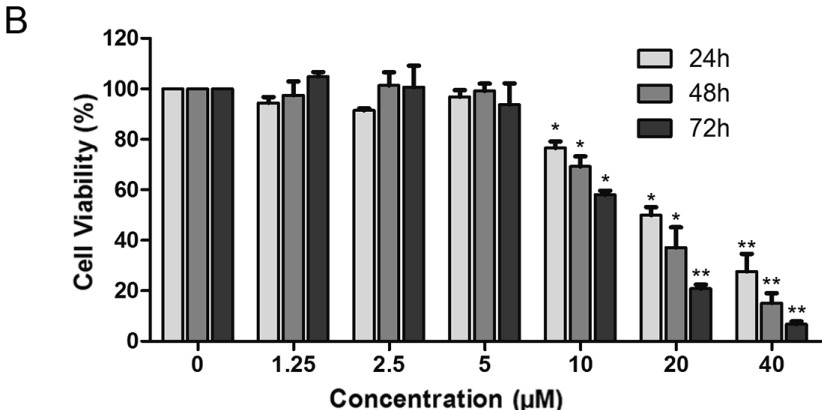

**Figure 1 The cytotoxicity of psoralidin against human lung cancer A549 cells.** (A) Chemical structure of psoralidin. (B) Cells were treated with series concentrations of psoralidin for 24, 48, or 72 h. The cell viability was measured by MTT assay, and the data were presented as means $\pm$ SD from three independent experiments. (C) Cells were treated with 5, 10, 20 and 30 μM psoralidin for 24 h, the LDH assay were performed. $*P < 0.05$ vs. Con and $**P < 0.01$ vs. Con. Con, control.

*Whelan & Ryan, 2003*). The major chemical constituents in *P. corylifolia L.* include coumarins, flavonoids and meroterpene phenols (*Song, Yang & Yuan, 2013*). Psoralidin (PSO, Fig. 1A), with coumarin structure, is one of the bioactive components. It is reported that PSO shows cytotoxic effects against some cultured human cancer cell lines (*Bronikowska et al., 2012*; *Kumar et al., 2010*; *Mar, Je & Seo, 2001*; *Pahari & Rohr, 2009*; *Yang et al., 1996*). In prostate cancer and HeLa cells, PSO enhanced TRAIL-mediated apoptosis (*Bronikowska et al., 2012*; *Szliszka et al., 2011*). PSO induced apoptosis in breast cancer cells by inhibiting NOTCH1 signaling (*Suman, Das & Damodaran, 2013*). It also inhibits TNF-mediated survival signaling in androgen independent prostate cancer cells (*Srinivasan et al., 2010*). However, its effect on autophagy, a nonapoptotic form of programmed cell death, remains to be clarified. Herein, we demonstrated that PSO is an anti-proliferative natural compound on human lung cancer A549 cells. It induces autophagy rather than apoptosis, which is triggered by increasing intracellular ROS generation.

## MATERIALS AND METHODS

### Reagents and cell culture

Psoralidin (>98%) was purchased from Chengdu Preferred Biotech Co. Ltd. (Chengdu, China). Dimethyl sulfoxide (DMSO), MTT, Hoechst 33342, monodansylcadaverine (MDC), propidium iodide (PI), 3-methyladenine (3-MA), Ac-DEVD-CHO, z-VAD-FMK, DAPI, CM-$H_2$DCF-DA, N-acetyl cysteine (NAC), Annexin V-FITC were from Sigma Aldrich (St. Louis, MO, USA). Cytotoxicity Detection Kit (lactate dehydrogenase, LDH) was obtained from Roche Diagnostics (Mannheim, Germany). Antibodies against LC3, Bcl-2, BAX, PARP, Caspase-3, Caspase-9 and GAPDH were purchased from Cell Signaling Technology (Beverly, MA, USA).

The human lung cancer cell line A549, obtained from American Type Culture Collection (ATCC, USA), was cultured in RPMI 1,640 (Gibco) supplemented with 10% (v/v) fetal bovine serum at 37 °C in a humidified atmosphere of 5% $CO_2$.

### MTT assay and LDH assay

Cells in the exponential growth phase were seeded in 96-well culture plates (5,000 cells per well), treated with various concentrations (1.25, 2.5, 5, 10, 20, 30 and 40 μM) of PSO for indicated time. After incubation, 20 μl MTT solutions (5 mg/ml) was added to each well and incubated for further 4 h. Then the supernatant was removed and the resulting crystals were dissolved in DMSO. The absorbance of each well was measured using a microplate reader (PerkinElmer, USA) at 570 nm. The cell viability was calculated by the formula: cell viability (%) = (average absorbance of treated groups/average absorbance of control group) × 100%. A commercial cytotoxicity detection kit was used to evaluate the LDH release from cells after treatment with different concentrations of PSO according to the manufacturer's protocol.

### Cell cycle analysis

After treated with PSO, cells were harvested and washed with cold phosphate buffer saline (PBS), and were fixed with 70% ethanol overnight at −20 °C. The fixed cells were then washed twice with cold PBS, and the supernatant was removed. Cells were stained with PI staining solution (10 μg/ml RNase A and 50 μg/ml PI) at 37 °C for 30 min in dark. The cell cycle distribution was analyzed using a flow cytometry provided with the Cell-Quest software (Becton Dickinson, USA).

### Apoptosis detection

Hoechst 33342 staining and Annexin V-FITC staining were performed to detect apoptosis. For Hoechst 33342 staining, A549 cells were washed with PBS and stained with Hoechst 33342 (1 μg/ml in PBS) at room temperature for 20 min, the fluorescence was observed by a fluorescence inverted microscopy. For Annexin V-FITC staining, the treated cells were collected, washed and then stained with Annexin V-FITC at room temperature for 15 min, the percentage of apoptotic cells were analyzed by flow cytometry.

## MDC staining and immunofluorescence

The autophagic activity was evaluated using MDC staining and immunofluorescence for LC3-II by fluorescence microscopy as described previously (*Zhang et al., 2013*). In brief, the treated cells were incubated with 0.05 mM MDC for 15 min in the dark, then washed with PBS twice and immediately analyzed by a fluorescence inverted microscopy. For immunofluorescence, the treated cells were fixed with 4% paraformaldehyde and blocked with 2% BSA for 30 min, incubated with primary antibody against LC3-II at 4 °C overnight, then washed with PBS twice and incubated with fluo-conjugated secondary antibody at room temperature for another 1 h. The nuclei were stained with DAPI and the stained cells were observed under a fluorescence inverted microscope using filter set for FITC and DAPI.

## Western blot analysis

After treatment with PSO, the cells were washed with cold PBS and lysed with RIPA lysis buffer (Santa Cruz, USA) to extract the total proteins. The concentrations of the total proteins were determined by Pierce BCA protein assay kit (Thermo Scientific, USA). Equivalent amounts of total proteins from each sample were separated by SDS-PAGE, and then transferred onto a PVDF membrane. After blocking with PBS containing 0.1% Tween-20 and 5% nonfat milk for 1 h, the transferred membrane was incubated with a specific primary antibody at 4 °C overnight, followed by incubation with the corresponding secondary antibody. Washing the membrane and specific protein bands were visualized using an ECL Advanced Western Blot detection Kit, the densitometric analysis of bands was performed using the Quantity-One Software.

## Determination intracellular ROS production

Intracellular ROS production was measured by flow cytometry analysis using $CM-H_2DCF-DA$ as fluorescence probe. After PSO treatment, cells were washed and incubated with $DCFH_2-DA$ (5 μM) for 30 min at 37 °C in the dark. The stained cells were analyzed by flow cytometry.

## Statistical analysis

Data are expressed as the means ± SD from at least three independent experiments. The differences between groups were analyzed by one-way ANOVA with Tukey's posthoc tests, significance of difference was indicated as $*P < 0.05$ or $**P < 0.01$.

## RESULTS

### The cytotoxicity of PSO in A549 cells

The effects of PSO in A549 cells were detected using MTT assay. As shown in Fig. 1B, the inhibition of PSO in A549 cell proliferation was exhibited both in time- and concentration-dependent manners. The calculated $IC_{50}$ after 24, 48 and 72 h treatment were 19.2, 15.4 and 11.8 μM respectively. Furthermore, after treatment with varies concentrations of PSO for 24 h, the LDH release from A549 cells was increased in a concentration-dependent manner (Fig. 1C).

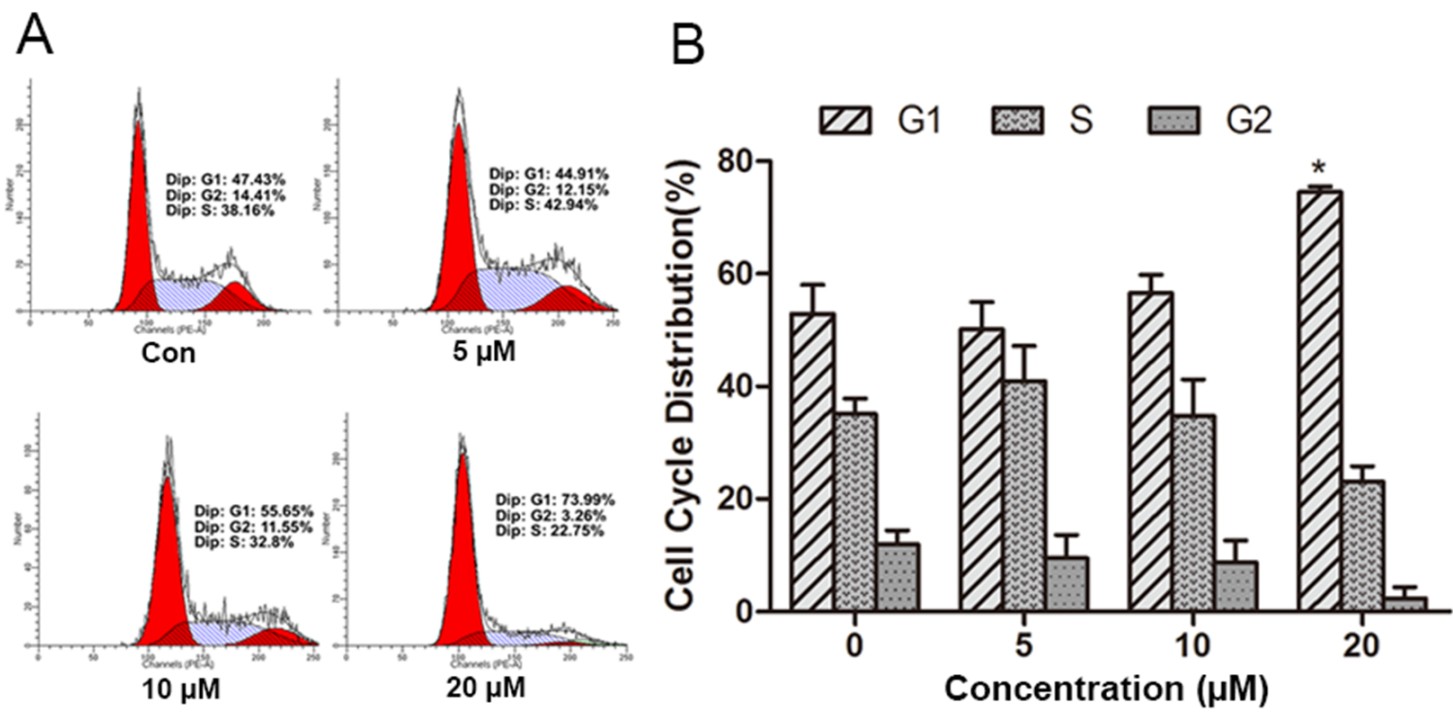

**Figure 2 Effect of psoralidin in cell cycle distribution of A549 cells.** (A) Cells were treated with 5, 10 and 20 μM psoralidin for 24 h, and then stained with propidium iodide. The DNA content was measured by flow cytometry. (B) The cell cycle distributions were analyzed and presented as mean ± SD of three independent experiments. *$P < 0.05$ vs. Con. Con, control.

## PSO induced cell cycle arrest at G1 phase

To measure the underlying mechanism responsible for the anti-proliferation effect of PSO in A549 cells, cell cycle distribution was detected by flow cytometry analysis of DNA content using PI staining. As shown in Fig. 2, PSO in concentrations of 10 and 20 μM induced an increase in the percentage of cells in G1 phase. Compared with the control group, 20 μM PSO treatment groups showed significant increase ($P < 0.05$) in the proportion of G1 phase cells (Fig. 2B).

## PSO showed little effect on apoptosis in A549 cells

Hoechst 33342 staining and flow cytometry assay using Annexin V-FITC staining were performed to determine whether apoptosis was induced by PSO. As shown in Fig. 3A, both the untreated and treated cells had regular and round-shaped nuclei and the characteristic morphological changes of apoptosis were not observed. Flow cytometry analysis using Annexin V-FITC staining showed that treatment A549 cells with different concentrations of PSO did not alter the percentage of Annexin V-FITC positive cells compared with control group (Fig. 3B). Furthermore, the expressions of some apoptosis-related proteins were analyzed by Western blot. After the treatment with PSO for 24 h, the expression of apoptosis-related proteins Bcl-2, BAX, Caspase-3, Caspase-9 and PARP did not alter significantly (Fig. 3C). These results suggest that PSO might not induce apoptosis in A549 cells.

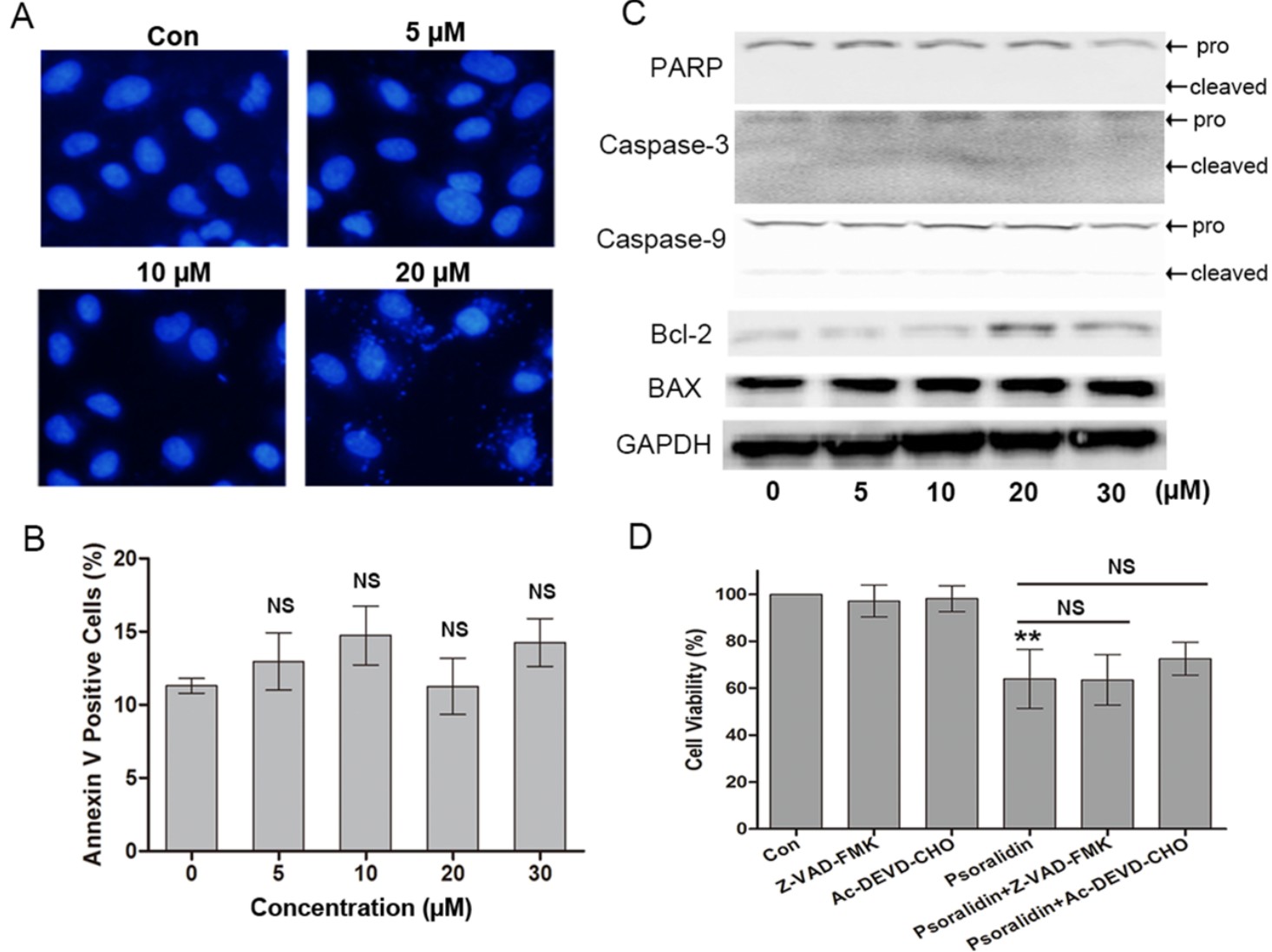

**Figure 3 Psoralidin showed little effect on apoptosis in A549 cells.** (A) Fluorescent staining of nuclei in A549 cells by Hoechst 33342. (B) Flow cytometry assay to detect apoptosis in A549 cells using Annexin V staining. There is no significant difference among control group and psoralidin treated groups. NS, no significant vs. Con. (C) The expression of apoptosis related proteins were analyzed by Western blot in A549 cells receiving psoralidin treatment. (D) The cell viability was measured by MTT assay after treatment with psoralidin in the absence or presence of Z-VAD-FMK or Ac-DEVD-CHO. **$P < 0.01$ vs. Con; NS, no significant vs. psoralidin only-treated group. Con, control.

In addition, the pancaspase inhibitor Z-VAD-FMK and caspse-3 inhibitor AC-DEVD-CHO were used to confirm the effect of PSO in A549 cells. As shown in Fig. 3D, Z-VAD-FMK and AC-DEVD-CHO failed to protect A549 cells from death caused by PSO. These results collectively indicate that the cell death induced by PSO is in a caspase-independent non-apoptotic manner.

## PSO induced autophagic cell death in A549 cells

To test the autophagic activity induced by PSO, MDC staining was performed first. As shown in Fig. 4A, the control cells showed faint fluorescence, while cells treated with

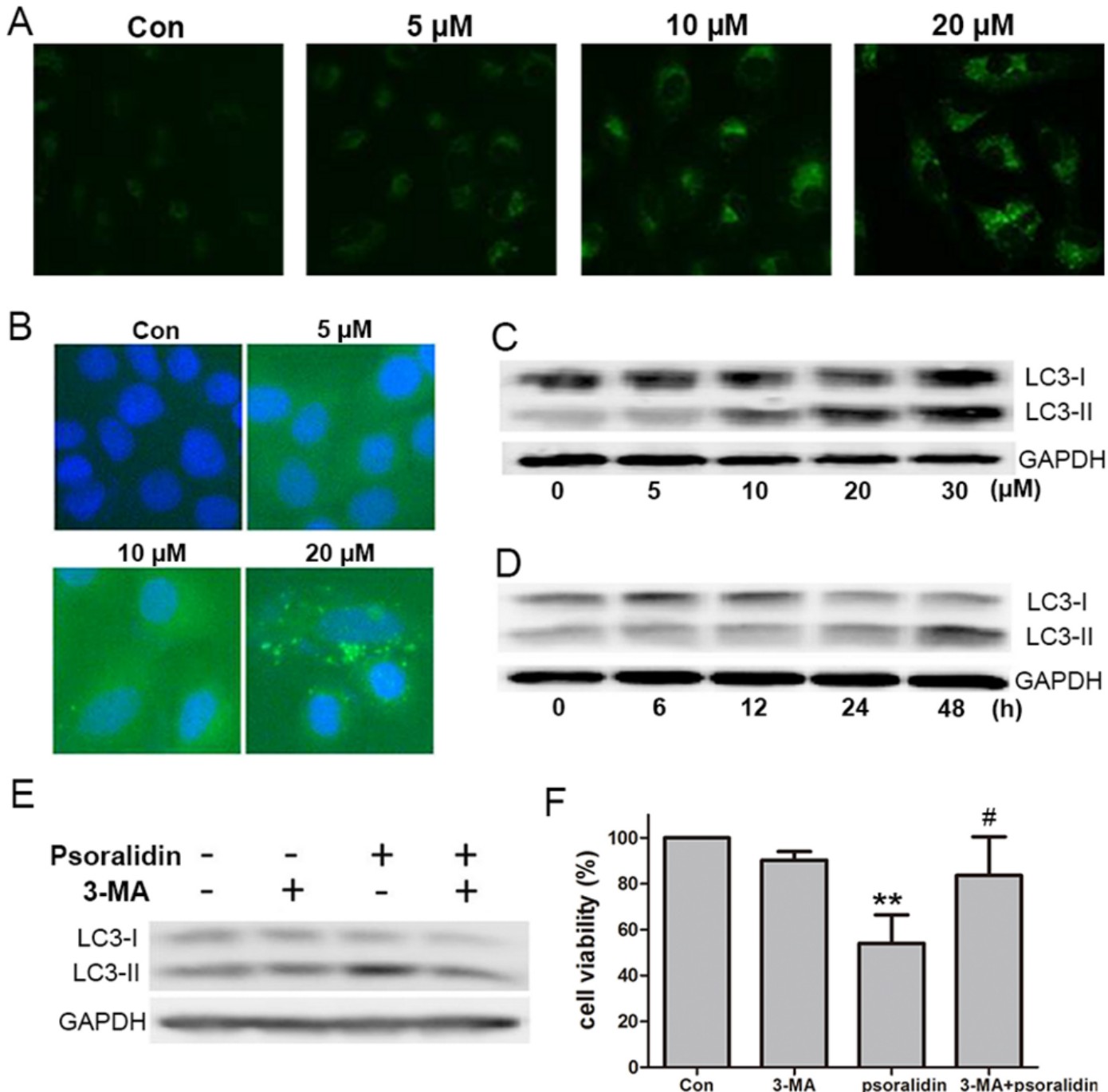

**Figure 4 Psoralidin induced autophagic cell death in A549 cells.** (A) Cells were stained with MDC after psoralidin treatment. (B) Immunoflueres-cence analysis of endogenous LC3 in psoralidin-treated A549 cells. (C) Cells were treated with indicated concentrations of psoralidin for 24 h. The expression of LC3 was determined by Western blot. (D) Cells were treated with 20 μM psoralidin for 6, 12, 24 or 48 h, expression of LC3 was determined by Western blot. (E) Cells were treated with psoralidin, 3-MA or a combination of both, expression of LC3 was analyzed by Western blot. (F) The cell viability was measured by MTT assay after treatment with psoralidin in the absence or presence of 3-MA. **$P < 0.01$ vs. Con, #$P < 0.05$ vs. psoralidin only-treated group. Con, control.

PSO accumulated MDC into granular structures of high fluorescence intensity. The immunofluorescence for LC3-II was executed to further examine the formation of autophagic vesicles. Compared with the untreated cells, A549 cells treated with PSO exhibited increases in both the number and size of LC3-II-postive puncta (Fig. 4B). Western blot analysis for LC3 showed a remarkable increase of LC3-II in response to PSO treatment both in concentration- and time-dependent manners (Figs. 4C and 4D). Furthermore, this increase of LC3-II could be blocked in the presence of autophagy inhibitor 3-MA (Fig. 4E). In addition, pretreatment with 3-MA significantly prevented cell death induced by PSO (Fig. 4F). Taken together, these results suggest that PSO induces autophagic cell death instead of apoptosis in A549 cells.

### PSO induced ROS generation and NAC reversed PSO-induced autophagy and cell death

As shown in Figs. 5A and 5B, PSO induced ROS generation in a concentration-dependent manner in A549 cells. NAC pretreatment efficiently attenuated PSO induced elevation of ROS levels. Meanwhile, pretreatment with NAC prevented the increase of LC3-II and cell death in response to PSO (Figs. 5C and 5D). These results suggested that the production of ROS plays a key role in the PSO-induced autophagy and subsequent cell death.

## DISCUSSION

PSO is a natural product isolated from *Psoralea corylifolia L.* and is an important medical herb prescribed in traditional Chinese medicine. Previous reports demonstrated that PSO showed significant antibacterial (*Khatune et al., 2004*), antidepressant-like properties (*Yi et al., 2008*) and antioxidant (*Xiao et al., 2010*) activities. It also showed inhibitory effects on protein tyrosine phosphatase 1B *in vitro* (*Kim et al., 2005*), induction of quinone reductase activity (*Lee, Nam & Mar, 2009*), and inhibitory effects on LPS-induced iNOS expression (*Chiou et al., 2011*). Furthermore, it was found that PSO could act as an ER agonist (*Liu et al., 2014*) and was a dual inhibitor of COX-2 and 5-LOX (*Yang et al., 2011*). Herein, the anti-proliferative effect of PSO was investigated.

Previous studies showed that PSO inhibited the proliferation of SNU-1 and SNU-16 gastric carcinoma cell lines (*Yang et al., 1996*), HT-29 colon and MCF-7 breast human cancer cell lines (*Mar, Je & Seo, 2001*). In the present study, we showed that PSO inhibited proliferation on human lung cancer A549 cells with $IC_{50}$ at the range of 10–20 μM, which suggested PSO possesses a potential cytotoxicity to cancer cells. This was further confirmed by PSO induced LDH release in the culture medium. The effect of PSO on cell showed that PSO induced an increase in the percentage of cells in G1 phase, suggesting PSO could cause the cell cycle arrest.

Apoptosis, a physiological process of programmed cell death, is one of the major types of cell death caused by most chemotherapeutics. It has been reported that PSO enhanced TRAIL-induced apoptosis in HeLa cells (*Bronikowska et al., 2012*) and prostate cancer cells (*Szliszka et al., 2011*). In the present study, the effect of PSO on apoptosis in A549 cells was detected using Hoechst 33342 staining by fluorescence microscopy and Annexin V-FITC

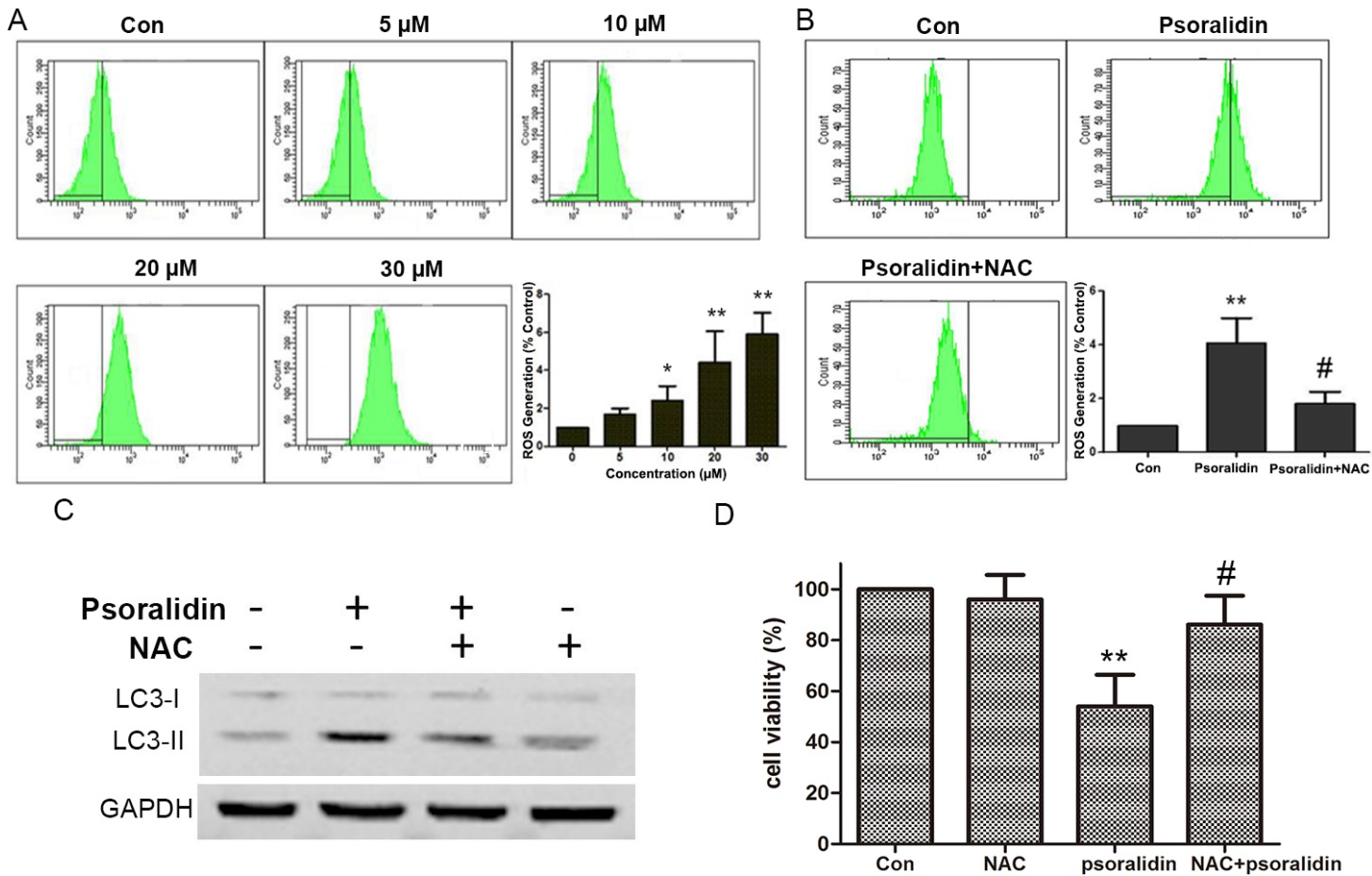

**Figure 5 Psoralidin induced ROS generation and NAC reversed psoralidin-induced autophagy and cell death.** (A) Cells were treated with indicated concentrations of psoralidin, and the intracellular ROS generation was detected as described in Materials and methods. *$P < 0.05$ vs. Con and **$P < 0.01$ vs. Con. (B) Cells were treated with 20 μM psoralidin in the absence or presence of NAC. The intracellular ROS generation was detected, and data represent the means ± SD from three independent experiments. **$P < 0.01$ vs. Con, #$P < 0.05$ vs. psoralidin only-treated group. (C) Cells were treated with 20 μM psoralidin in the absence or presence of NAC. The expression of LC3 was analyzed by Western blot. (D) The cell viability was measured by MTT assay after psoralidin with treatment in the absence or presence of NAC. **$P < 0.01$ vs. Con, #$P < 0.05$ vs. psoralidin only-treated group. Con, control.

staining by flow cytometry. In addition, the expressions of apoptosis-related proteins such as Bcl-2, Bax, Caspase-3, Caspase-9 and PARP were examined by Western blot. There was no significant nuclear changes in Hoechst 33342 staining were observed. Furthermore, no significant differences in the percentage of Annexin V-FITC positive cells and apoptosis-related protein expression between the control groups and PSO treated groups. In addition, the pancaspase inhibitor Z-VAD-FMK and caspse-3 inhibitor AC-DEVD-CHO failed to protect A549 cells from PSO induced death. Taken together, these results indicated that apoptosis might play a minor role in PSO induced cell death in A549 cells.

Autophagy, a cellular process responsible for the degradation of cytoplasmic components through an autophagosomal-lysosomal pathway, has been implicated to play a key role in cancer initiation and progress (*Janku et al., 2011*). A panel of natural products has been identified to induce both apoptotic and autophagic cell death. PSO showed

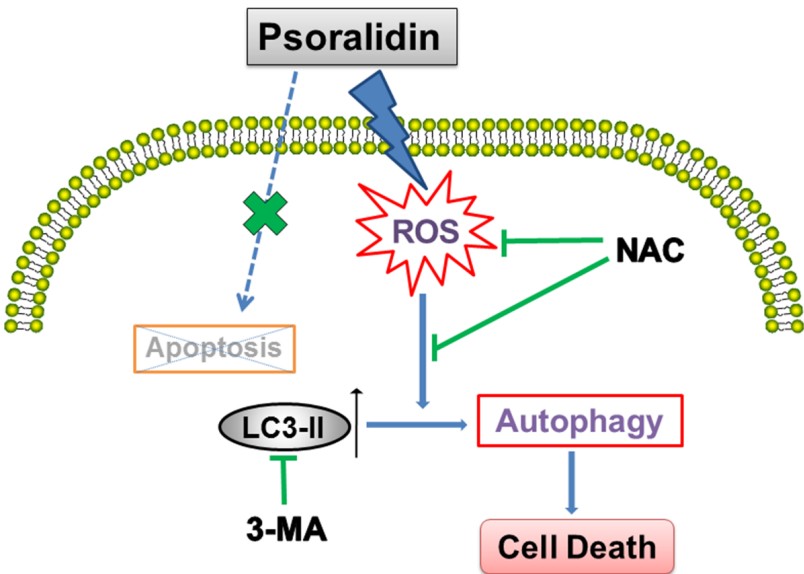

**Figure 6** Schematic diagram illustrates the underlying mechanism of psoralidin-induced cell death in A549 cells.

cytotoxicity to A549 cells but did not induce apoptosis. Therefore, we detected whether PSO induced autophagic cell death in A549 cells. MDC is an acidotropic dye that labels late stage autophagosomes or autophagic vesicles. During autophagosome formation, LC3-I is incorporated into autophagosome membranes mediated by Atg3 and Atg7, which results in the conversion of cytosolic LC3-I into membrane-bound form LC3-II (*Tanida, Ueno & Kominami, 2004*). Thus, both the MDC staining and the expression ratio of LC3-I to LC3-II provides simple indicators for autophagy activity and has been widely used to monitor the autophagic activities (*Klionsky et al., 2012*). Present results showed that PSO induced MDC staining, punctate staining dots for LC3-II, and increased protein expression ratio of LC3-II/LC3-I suggest that PSO caused autophagy in A549 cells. Furthermore, pretreatment with 3-MA, an autophagy inhibitor, reversed PSO induced significantly protein expression ratio of LC3-II/LC3-I and cell death induced by PSO. Taken together, these results suggested that PSO induced autophagic cell death instead of apoptosis in A549 cells. To the best of our knowledge, this is the first report on the effect of PSO on autophagy.

Accumulated evidence suggested that ROS, a class of important multifaceted signaling molecules, is implicated in a variety of cellular programs including autophagy (*Azad, Chen & Gibson, 2009*; *Dewaele, Maes & Agostinis, 2010*; *Scherz-Shouval & Elazar, 2011*). Though PSO showed antioxidant activities in scavenging DPPH and ABTS free radicals in *ex vivo* models (*Wang et al., 2013*; *Xiao et al., 2010*), it greatly induced ROS generation that resulted in the growth inhibition and promote epithelial-mesenchymal transition in prostate cancer cells (*Das, Suman & Damodaran, 2014*). We also found that PSO significantly induced DCF fluorescence in A549 cells suggesting the increase of intracellular ROS generation. NAC pretreatment could significantly reverse PSO induced autophagic biomarkers and cell death suggested that ROS mediated PSO-induced autophagy and subsequent cell death.

In summary, as shown in Fig. 6, our results show that PSO is a potential cytotoxic natural compound. Instead of apoptosis, it induces ROS-triggered autophagy which inhibits the growth of human lung cancer A549 cells.

### Funding

The present study was supported by the Research Fund of the University of Macau (No. MYRG118(Y1-L4)-ICMS13-CXP, MRG007/CXP/2013/ICMS) and the grants from Science and Technology Development Fund of Macau Special Administrative Region (No. 021/2012/A1). The funders had no role in study design, data collection and analysis, decision to publish, or preparation of the manuscript.

### Grant Disclosures

The following grant information was disclosed by the authors:
Research Fund of the University of Macau: MYRG118(Y1-L4)-ICMS13-CXP, MRG007/CXP/2013/ICMS.
Science and Technology Development Fund of Macau Special Administrative Region: 021/2012/A1.

### Competing Interests

The authors declare there are no competing interests.

### Author Contributions

- Wenhui Hao performed the experiments, analyzed the data, contributed reagents/materials/analysis tools, prepared figures and/or tables.
- Xuenong Zhang performed the experiments, analyzed the data, contributed reagents/materials/analysis tools, wrote the paper, prepared figures and/or tables.
- Wenwen Zhao performed the experiments, contributed reagents/materials/analysis tools, wrote the paper.
- Xiuping Chen conceived and designed the experiments, analyzed the data, wrote the paper, reviewed drafts of the paper.

### Supplemental Information

Supplemental information for this article can be found online at http://dx.doi.org/10.7717/peerj.555#supplemental-information.

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
