# Peer review of "Psoralidin induces autophagy through ROS generation which inhibits the proliferation of human lung cancer A549 cells"

_PeerJ, doi:10.7717/peerj.555_

## Round 0.1 · original submission · Major Revisions

· Academic Editor

Major Revisions

Dear Dr Chen,
Your manuscript have been revised by two indepent expert in the field. Thought they found the manuscript interesting, both reviewers raised questions about the necessity of using a different cell line (one of the reviewers suggest to be used lung cells) to clarify why in some cell lines apoptosis was found and in your work only autophagy. Note that this point can be critical in the re-review of your revised manuscript.

Reviewer 1 ·

Basic reporting

The preceding articles about the effects of psoralidin on cancer cells have shown that apoptosis occurs by this chemical, but this study has shown that autophagic death is the main cause of cell growth inhibition. It is necessary to explain the reasons (except for difference in cell line used) why different results between this study and the other studies dealing with psoralidin occured. To confirm the essence of this study, it is suggested to show similar results using another lung cancer cell line.
In Materials and methods, please complete the material information with company name, city name and nation name. Many materials afford incomplete information.
Please correct usage for the name of the Psoralea corylifolia L
P3 line 36: Psoralea corylifolia L. (Leguminosae) should be replaced by Psoralea corylifolia Leguminosae (L. ). And, Psoralea corylifolia L. should be used throughout the manuscript in Italic font.

Experimental design

No comments

Validity of the findings

Regarding to Fig. 1B and 1C, this referee supposes that Fig.1C is redundant and unnecessary. But, reading the legend for Fig. 1C, this referee is very afraid that the Fig. 1B must have been made from single series of experiment. Please delete Fig. 1C, and please replace Fig. 1B by revised Fig. 1B after properly repeated experiments.
Regarding to Fig. 4, this referee cannot understand the presentaion of Western blotting. Especially, why don't the authors show cleaved form caspase 3 and full view of caspase 9 expression pattern. Anyone cannot tell apoptosis status from these bands shown.

Additional comments

This manuscript written by W. Hao, et al. has shown that psoralidin, content of Psoralea corylifolia L., induces autophagy, not apoptosis, to inhibit cell proliferation of a human lung adenocarcinoma cell line. In general, this study is well organized, concise and seems to present reasonable conclusion. However, some problems are found in this manuscript, and the referee suggests that revision is necessary before acceptance for publication in PeerJ.

Reviewer 2 ·

Basic reporting

No Comments.

Experimental design

The manuscript report a series of carefully done experiments that are aimed to explore the mechanisms underlying the anti-cancer effects of PSO. The authors found that PSO inhibited the proliferation of A549 cells through autophagy rather than apoptosis. The major limitation of this study is that all the results are based on only one human cancer cell line A549. I suggested the authors extend their observations to other cancer models.

Validity of the findings

This manuscript describes mechanistic studies of Psoralidin (PSO), a natural furanocoumarin with anti-cancer properties. The authors concluded that PSO inhibited the proliferation of A549 cells through autophagy rather than apoptosis, and these effects were mediated by inducing ROS production.
General comments:
ROS generation has been suggested to be a major mechanism underlying the anti-cancer effects of various agents. ROS could induce many kinds of cell death, including apoptosis and autophagy. The authors showed that PSO failed to induce cell death via apoptosis. Why PSO-induced ROS generation leads to cell autophagy rather than apoptosis in this cancer cell line?
Specific comments:
1. Similar results were demonstrated in Fig. 1B and 1C, the author could omit one. I consider that data presented as 1B is adequate.
2. Caspase inhibitors Z-VAD-FAM or Ac-DEVD-CHO was used in Fig.3 to assess whether PSO induced caspase-dependent apoptosis in A549 cells. Whether the inhibitors used in the experiment have cytotoxicity?The author should show the effects on cell viability with treatment of inhibitors alone.
3. The quanlity of Fig.4B is poor so that changes in number and size of LC3-II-positive puncta are not evident.
4. The author found pretreatment of 3-MA reversed PSO induced ratio of LC4-II/LC-I and cell death. I suggested other inhibitors such as chloroquine could be used to confirm these effects. The author should show the effects on cell viability with treatment of 3-MA alone in Fig.5F.

---

## Round 0.2 · accepted · Accept

· Academic Editor

Accept

Note that your statement:

"NAC, a ROS scavenger, pretreatment could significantly reverse PSO induced autophagic biomarkers and cell death suggested that ROS mediated PSO-induced autophagy and subsequent cell death. "

has to be corrected or supported by proper data. Indeed, NAC is a precursor of GSH. Clarify this and thank you.

Reviewer 1 ·

Basic reporting

No comments

Experimental design

No comments

Validity of the findings

No comments

Additional comments

This referee considers that the revised manuscript has been much improved.

Reviewer 2 ·

Basic reporting

No comment

Experimental design

No comment

Validity of the findings

No comment

Additional comments

Some suggested changes have been applied. The response to the comments are reasonable. I recommend the manuscript for publication.